# Rethinking infrastructure design from component failure to systemic resilience

Sam Dulin [1,2], Stergios-Aristoteles Mitoulis[3,4], Alexandre Bredikhin[5], Eric Treyz [1,6], Billy Leung[7], Jeffrey Dykes[7], Owen Karpeles[7], Shreeya Gurav[7], Alex Karhunen[7] & Igor Linkov [1,8] ✉

Bridge design typically uses load-based design criteria focused on risk thresholds from engineering practice and standards, overlooking cascading effects on connected infrastructure and regional economies. We argue for a systems-based design that balances risk reduction with resilience − the capacity to recover from disruptions. Using the Francis Scott Key Bridge collapse as a case study, we estimate economic impacts assuming impact on local transportation networks only as well as integrating cascading failures on surrounding infrastructure (e.g., closure of the Port of Baltimore), employing the regional economic model TranSight. Results show combined bridge-and-port disruptions produce substantially larger losses in GDP, employment, disposable income, and labor force, with some indicators not recovering until 2040. The Baltimore region exhibits lower resilience to compounding shocks, highlighting the need for a resilience-based framework that considers interconnected infrastructure. We conclude infrastructure design must move beyond component-focused risk criteria toward an explicit, quantifiable resilience framework.

The recent failure of the Francis Scott Key Bridge in Baltimore (Key Bridge), in which debris blocked the path of routes to the port and impacted port operations and regional transportation networks, underscores the profound impact that infrastructure disruptions can have on transportation and interconnected economic and social networks. The cost of rebuilding the bridge is an estimated 1.7 billion USD[1], but the cost of the collapse has been greatly compounded by the approximately 15 million USD dollars lost every day during the nearly 11-week port closure that resulted[2]. Furthermore, there is the ongoing impact to personal incomes due to the large number of port workers impacted by the port closures[3]. Even with the port reopening relatively quickly within a 11-week time frame, operability is still at about only 85% as of September, according to the Maryland Port Administration[4]. A design paradigm focused on cascading

consequences and resilience rather than traditional risk measures alone may be required.

The Key Bridge collapse is not an isolated event (see Supplement for more examples), it is part of a historical pattern that underscores the complex interdependencies with modern infrastructure systems (refs. 5–11 Bridge failures not only disrupt local and regional mobility but can also set off cascading effects across multiple systems, including supply chains, public services, and broader economic structures[12–15], These effects can extend into the medium and long term, with consequences that are often disproportionate to the initial cause.

The bridge design guidelines call for mitigating risks primarily through load-based approaches and neglect the consequences of the collapse and potential cascading effects, and has a qualitative guidance

[1]Environmental Laboratory, US Army Engineer Research and Development Center, 696 Virginia Rd, Concord, MA, USA. [2]Credere Associates, Westbrook, ME, USA. [3]The Bartlett School of Sustainable Construction, University College London, London, UK. [4]MetaInfrastructure.org, London, UK. [5]U.S. Army Corps of Engineers, Pittsburgh, PA, USA. [6]Bates College, 2 Andrews Rd, Lewiston, ME, USA. [7]Regional Economic Models Inc., 433 West St, Amherst, MA, USA. [8]Department of Engineering and Public Policy, Carnegie Mellon University, Wean Hall, 5000 Forbes Ave, Pittsburgh, PA, USA. ✉e-mail: Igor.Linkov@usace.army.mil

for minimizing losses. For example, a substantial portion of bridge failure - approximately 14% - can be attributed to ship collisions, highlighting a tangible risk to key bridge infrastructure. Thresholds are established to withstand specific collision scenarios, and these thresholds are translated in specific engineering designs. For critical bridges, the assessment of the risk of vessel collisions with bridge piers integrates in a bottom-up approach (AASHTO adapted Method II of the Guide specifications[16,17]), vessel traffic analysis, structural assessments, probability calculations, and risk distribution to deliver a risk management plan for bridge piers in navigable channels. For "critical bridges" (this term is not well defined in the Guide), the Guide identifies a risk target for collapse of 1 in 10,000 years (i.e., Annual Frequency of Collapse AF = 0.0001) irrespective of consequences.

This risk-based approaches to bridge design does not consider resilience. Resilience of a system is defined by the National Academy of Sciences (NAS) as its ability "to plan and prepare for, absorb, respond to, and recover from disasters and adapt to new conditions"[18]. Recovery and consequences to connected infrastructure, such as ports, are not even considered in the design guidelines.

It is not only critical that bridge design move from risk toward resilience, but that the resilience quantification measures more than the resilience of a region to traffic disruptions, which is one shortcoming of existing methodologies to quantify the resilience of bridge disruptions. While probabilistic and large-span bridge traffic load models, such as those developed by Yin et al.[19] and Testa et al.[20], are valuable for assessing structural resilience, they do not consider the economic ripple effects that emerge from interconnected systems, which are central to our resilience analytics. For example, the authors in refs. 21,22 have assessed bridge closures' impact on traffic flow and highway networks in their regional communities. While these studies provide insight into the impact of bridge failures, they do not consider the impact of the bridge collapses on interconnected infrastructure systems.

Other approaches to quantifying bridge resilience are load-based approaches (e.g., refs. 23–25), which do not account for the extensive social and economic losses that may follow after a bridge collapse. In situations when interdependency is low, such as bridges in rural areas, a failure might result only in direct repair and replacement costs. Yet, within highly interdependent infrastructure systems inside or near densely populated areas, the scale and scope of distribution may be much higher.

Clearly, not all bridges should be built equal, and so the impact of the bridge collapse on the regional economy should be considered. To illustrate the magnitude of the problem, this paper uses regional economic modeling to assess the impacts of a Francis Scott Key Bridge and/or Port of Baltimore shutdown. Based on our observation, we argue here that a full economic and systems resilience analysis should be included in the bridge tender and approval, justifying the selection of the bridge risk of collapse and design of additional measures, as opposed to a risk-based, binary selection which leads to the design of an either "critical" (AF = 0.0001) or "typical" (AF = 0.001) bridge. The economic resilience assessments used in this paper will clarify which designs could trigger tipping points, potentially leading to regional or national disasters, coined here as "vital to resilience" assets[26,27],

## Results

### Economic impacts

The simulation results reveal varying magnitudes of economic impacts across five infrastructure closure scenarios. In the most severe scenario, combining a four-year bridge closure with a 11-week port closure, total employment decreases by 23,968 workers in 2024, accompanied by a reduction of 6146 people in the labor force (Fig. 1). This scenario also results in a decrease of $2.237 billion in GDP (2017 dollars), a reduction in output of $3.953 billion (2017 dollars), and a decline in real disposable personal income of $1.691 billion (2017 dollars).

The combined 11-week closure of both facilities shows the second largest impacts, reducing employment by 15,829 workers, decreasing the labor force by 3724 people, and lowering GDP by $1.412 billion (2017 dollars). The isolated 11-week port closure generates the third

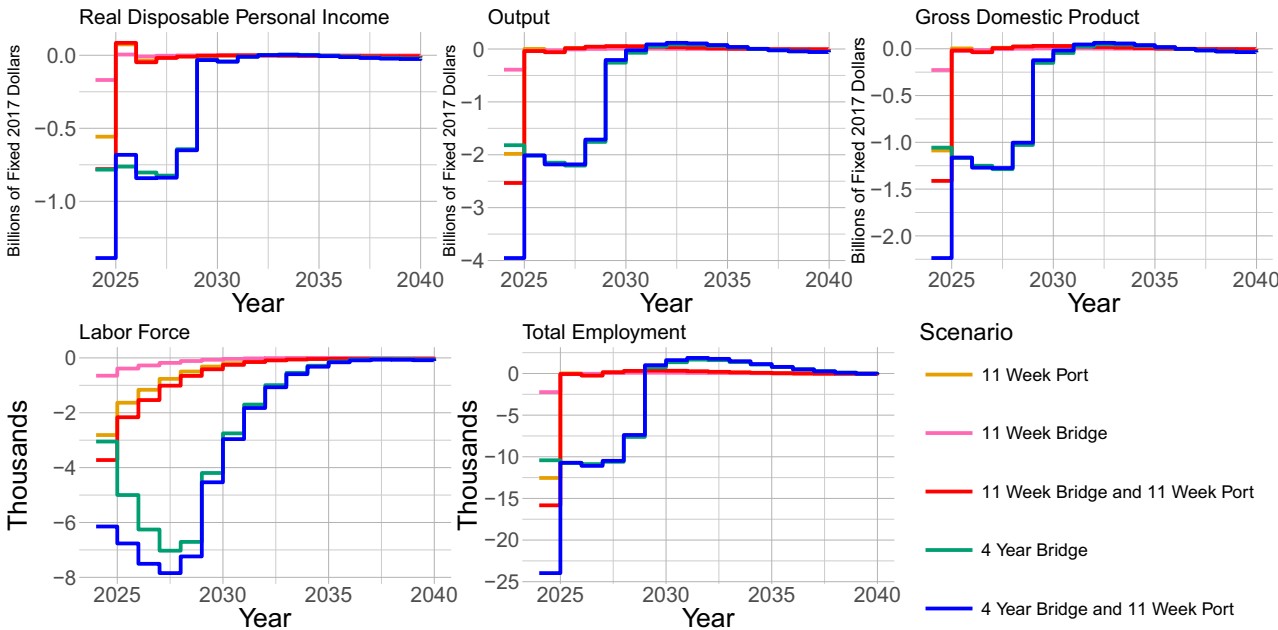

**Fig. 1 | Forecasted time series for five economic indicators—Gross Domestic Product (GDP), Total Employment, Labor Force, Output, and Real Disposable Income—under five distinct Key Bridge and/or Port of Baltimore closure scenarios.** Each plot represents a different variable, with step-function lines illustrating the annual impacts of the closure scenarios projected until 2040. These simulations provide a comparative view of how each closure scenario uniquely affects each economic indicator over time. Notably, the compounding closure scenarios (represented by the blue and red lines) result in greater impacts compared to their single-closure counterparts. We also note that in the combined 11-week scenario, most of the impact is due to the port, making it very close to the 11-week port closure scenario.

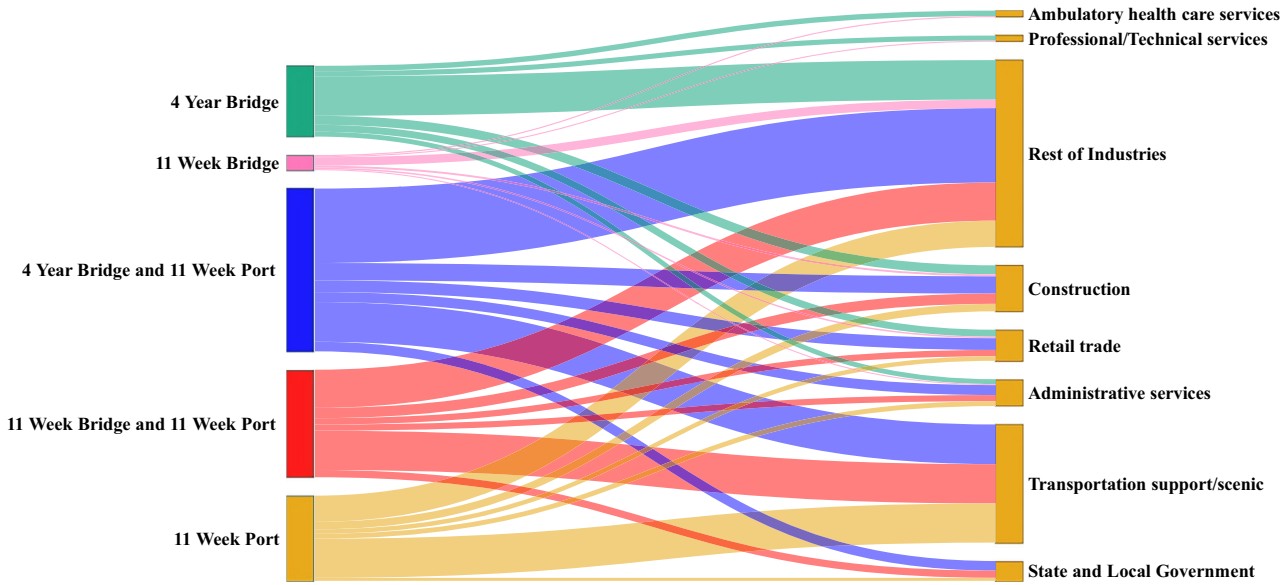

**Fig. 2 | Simulated 2024 impacts to employment in different sectors under different bridge/port shutdown scenarios.** The support activities for transportation/scenic transportation sector, which includes services such as marine cargo handling and ship repair, are impacted the most.

most severe impacts, with 12,545 fewer employed workers, a reduction of 2812 people in the labor force, and a $1.087 billion decrease in GDP (2017 dollars).

More moderate effects are observed in the 11-week bridge closure scenario, which results in employment declining by 2235 workers, the labor force decreasing by 650 people, and GDP reducing by $227 million (2017 dollars). Real disposable personal income shows similar patterns but with varying magnitudes across scenarios, ranging from $1.443 billion in losses in the combined four-year bridge and 11-week port scenario to $228 million in the 11-week bridge closure case.

Recovery patterns indicate that most economic indicators begin to improve by 2025, though labor force impacts persist through 2040 in the more severe scenarios. GDP and output generally show faster recovery than employment and labor force measures, typically approaching baseline levels by 2026–2027. Employment growth returns to positive territory by 2026–2027 in all but the most severe scenario.

Industry classifications follow the North American Industry Classification System (NAICS) framework[28]. The simulation of bridge and/or port shutdowns reveals differential impacts across economic sectors, with transportation-related industries experiencing the most severe employment losses, as can be seen in Fig. 2. Support activities for transportation, which include services such as marine cargo handling and ship repair, suffered the greatest losses across all scenarios. In the 11-week port closure scenario, this sector experienced a loss of 5770 jobs, while the combined 11-week bridge and port closure resulted in 5780 job losses. The extended 4-year bridge closure combined with the 11-week port closure produced virtually identical losses at 5780 jobs, indicating that bridge closure duration minimally compounds the port closure impact on transportation employment.

State and local government and construction sectors typically ranked second and third in terms of employment losses. In the most severe scenario (4-year bridge and 11-week port closure), state and local government lost 1410 jobs while construction lost 2550 jobs. Notably, the construction sector showed evidence of compounding effects: the 11-week port closure alone resulted in 1100 job losses, while the 4-year bridge closure alone caused 1330 job losses, and their combined impact (2550) exceeded the simple sum of individual effects. The aggregate impact on all remaining NAICS industries not explicitly examined showed substantial compounding effects, with

losses of 3860 jobs from port closure alone, 5820 from bridge closure alone, and 10,960 jobs when combined—exceeding the sum of individual impacts.

## Resilience
The scenario involving simultaneous closures of the bridge and port resulted in the lowest resilience scores across all economic variables (Fig. 3). This highlights the compounded impact of multiple infrastructure disruptions on the region. Notably, when both the bridge and port were closed, resilience was much lower compared to scenarios with either infrastructure closed individually.

Among the economic variables, the labor force displayed the lowest resilience score, with a notably extended recovery period following disruptions. For example, in the case of a bridge closure, the labor force took much longer to recover compared to other variables. This indicates a structural shift in workforce availability, which may stem from changes in population or workforce participation after the disruption.

Total employment, on the other hand, exhibited a much higher resilience score, recovering within one year across all disruption scenarios. Resilience scores for total employment were notably higher in bridge closure scenarios than in port closure scenarios, indicating that the bridge closure had a less severe impact on employment compared to the port closure of similar duration.

Other economic indicators, such as GDP and income, displayed resilience scores between 0.75 and 1, suggesting a relatively robust ability of these variables to recover from disruptions. The high resilience scores for these variables reflect Baltimore's economic strength and capacity to withstand shocks, as these indicators largely returned to pre-disruption levels within a short time frame.

## Compounding impacts
The compounding effects for two combined bridge and port closure scenarios can be seen in Fig. 4. In our analysis of these two scenarios—an 11-week simultaneous closure of the Francis Scott Key Bridge and Port of Baltimore, and a four-year bridge closure with a 11-week port closure—we observed a relatively larger nonlinear impact in the 11-week concurrent bridge and port closure scenario. The initial shock was marked by a positive multiplicative effect in the first year, suggesting a sudden, intensified disruption across many of Baltimore's economic indicators, including GDP, Output, Total Employment, and

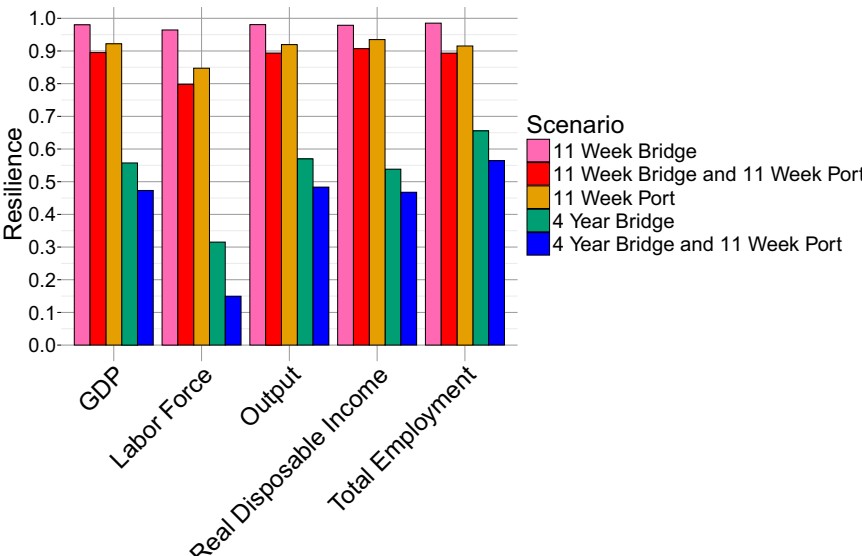

**Fig. 3 | Resilience scores for bridge/port scenarios.** For each economic indicator and closure scenario described in Fig. 1, this plot presents resilience scores, where a score of 1 represents perfect resilience and 0 reflects a worst-case response. Resilience here quantifies the capacity of each variable to withstand the economic shock introduced by each closure scenario. The scores highlight variations in resilience across GDP, Employment, Labor Force, Output, and Real Disposable Income, illustrating the relative stability or vulnerability of each indicator when faced with bridge or port disruptions. Note that the Baltimore region displays less resilience to compounding closure scenarios.

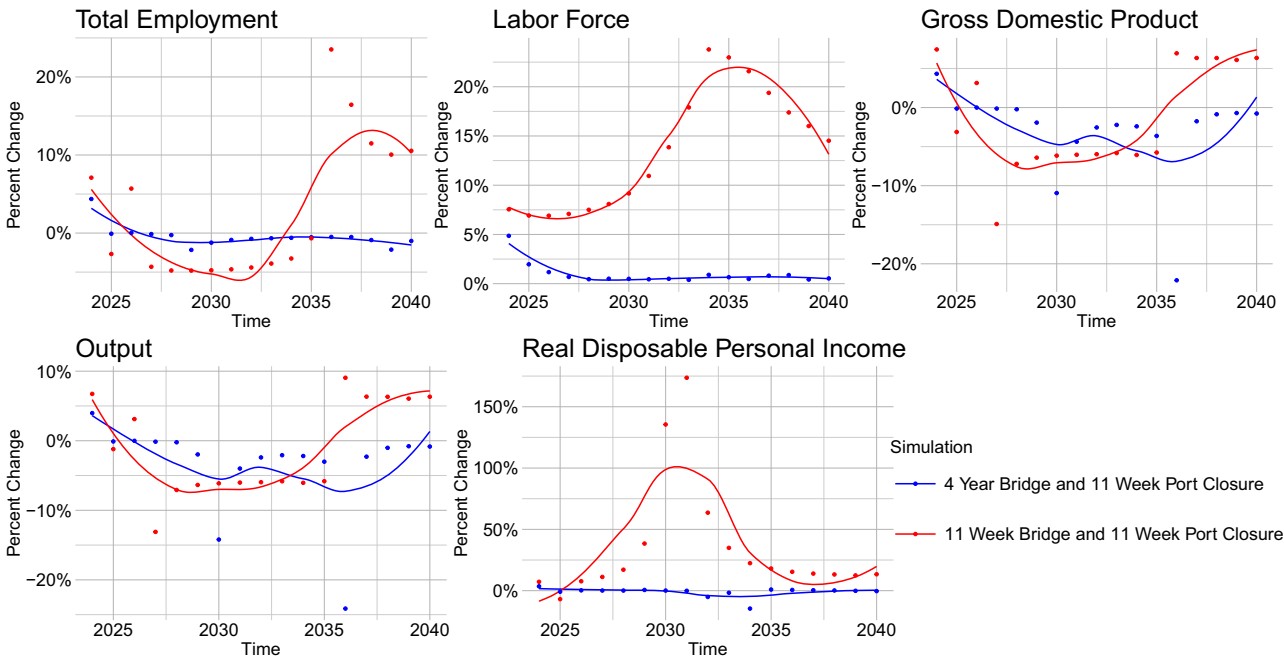

**Fig. 4 | Percent deviation from a linear projection for scenarios in which both the Key Bridge and Port of Baltimore experience closures, illustrating compounding effects.** Plots present forecasted percent changes annually from 2023 to 2040, with values near zero representing near-linear impacts and values diverging from zero indicating greater compounding effects. This view underscores the increasing or diminishing returns in economic impact when the infrastructure disruptions are combined, highlighting areas where compounded closures amplify or mitigate forecast deviations.

Labor Force. However, both scenarios displayed impacts that were relatively close to linear, with effects following predictable, additive patterns rather than escalating further. An exception to this trend was real disposable income.

For GDP and Output, the economy's return to a near-linear response indicates that essential economic activities, while initially strained, manage to regain equilibrium without exacerbating losses in unexpected sectors. Similarly, while the impact on employment is not perfectly linear, the impacts do not cascade in a way that could destabilize the labor market. Real Disposable Income (DPI) also stabilizes quickly to near linear effects for the four-year bridge closure with a 11-week port closure, while taking much longer to do so in the case of the 11-week concurrent bridge and port closure.

## Discussion
The integration of resilience quantification with regional economic modeling described above quantifies the economic and systemic impacts of infrastructure failures in detail. It also advances our

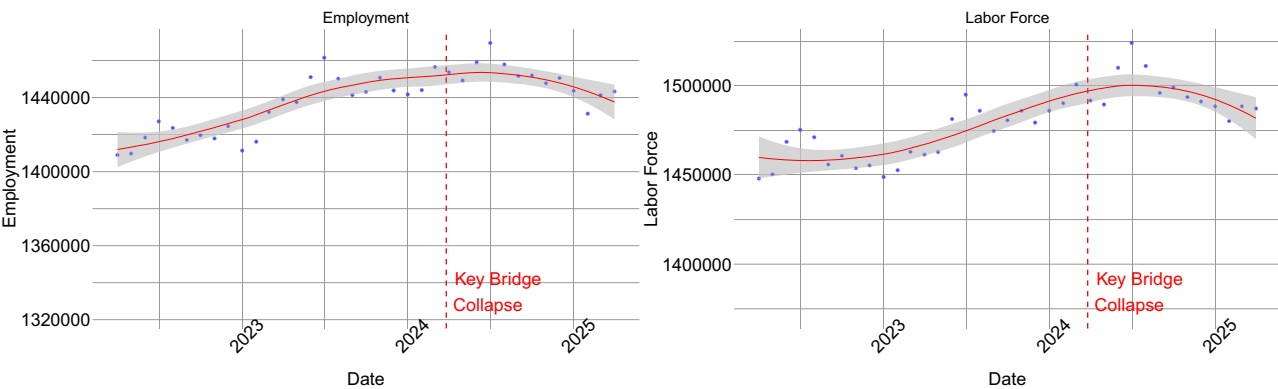

**Fig. 5 | Real economic data for the Baltimore-Columbia-Towson, MD Metropolitan Area.** The economic data points (blue dots) and fitted LOESS regression lines (red lines) with 95% confidence bands are included in this figure. The data is from ref. 38.

understanding of infrastructure's critical roles at the system, regional, and even national levels, providing a robust framework to quantify resilience against compounding impacts in order to prioritize investments and policy interventions based on both resilience and economic outcomes. The economic resilience findings discussed below demonstrate how single infrastructure disruptions can cascade across regional systems, emphasizing the critical need for frameworks that prioritize recovery and adaptability to safeguard societal stability and economic productivity.

Our analysis reveals a disparity between considering only direct impacts rather than both the direct and indirect impacts of the Key Bridge collapse, both in terms of short- and long-term impacts, as can be seen in Fig. 1. This is evidenced by the fact that the combined 11-week closure leads to job losses (15,829 workers) that exceed the sum of individual closure impacts (12,545 for port and 2235 for bridge alone).

The notably larger impacts of port closures compared to bridge closures of equal duration emphasize the port's crucial role as an economic hub. The 11-week port closure causes more than six times the employment impact of an 11-week bridge closure (12,545 versus 2235 jobs) and approximately six times the GDP impact ($1.087 billion versus $227 million). This asymmetry likely reflects the port's function in facilitating trade flows that cannot be easily rerouted, whereas bridge traffic may have alternative, if less efficient, routing options.

The divergence between output and employment recovery patterns reveals important labor market dynamics. While GDP losses show relatively quick recovery, employment and labor force impacts show more persistence. For example, in the combined four-year bridge and 11-week port scenario, even by 2035, the labor force remains below baseline, despite GDP recovering much earlier. This pattern suggests that workers who leave the labor force during the disruption may not readily return even after economic conditions improve.

The relative scales of impact across different economic measures provide insight into the nature of the disruption. A 11-week port closure, for instance, reduces GDP by $1.087 billion while affecting nearly 13,000 jobs, suggesting that the affected economic activity is relatively labor-intensive.

The results also suggest that combined disruptions (bridge and port shutdowns) resulted in the lowest resilience scores across all variables. This finding illustrates the importance of considering interconnected infrastructure vulnerabilities in design. While single disruptions had relatively moderate impacts on economic variables, the compounded effect of simultaneous disruptions demonstrates the greater reduction in regional resilience when multiple critical systems are impaired. This is particularly relevant in densely interconnected regions like Baltimore, where infrastructure components rely on one another to support economic functionality.

The contrasting resilience scores between labor force and total employment reveal important labor market dynamics. Employment recovery suggests that local businesses and employers were able to adapt to the initial shock and restore employment levels. However, the lagging labor force recovery indicates possible long-term shifts in population and workforce availability, suggesting that individuals may have relocated from the area or exited the workforce following the disruption. This underscores that resilience, as measured by total employment, may not fully capture long-term system-wide impacts on human capital, particularly if outmigration is contributing to the labor force's delayed recovery.

The pattern observed implies that the Baltimore economy experiences a sharp but contained impact under short-term, high-intensity disruptions, especially when key transportation and trade infrastructure are simultaneously shut down. While the economy's response to short-term disruptions is more nonlinear, the effects of longer disruptions tend to be more predictable, with a return to a linear pattern after the initial shock. This suggests that, while the initial impacts of a sudden, high intensity event (like a bridge and port closure) can be difficult to forecast, the economy can stabilize relatively quickly and resume a predictable pattern of recovery. However, the more predictable, linear response to longer disruptions doesn't necessarily indicate less vulnerability—rather, it shows that the economy is better able to absorb the shock over time without additional compounded losses.

The modeled economic impacts exceeded the observed real-world effects following the Francis Scott Key Bridge collapse, as evidenced by the relatively stable labor market conditions shown in Fig. 5. While the Baltimore-Columbia-Towson metropolitan area experienced a gradual decline in employment and labor force participation in the months following the FSK bridge collapse, the observed impacts were far less severe than model predictions. This divergence reflects several key limitations in translating model outputs to real-world outcomes.

The primary explanation for this discrepancy lies in the substantial federal and state interventions that occurred following the collapse, which are not captured in the REMI model's equilibrium framework. Programs such as Maryland's PORT Act and federal emergency funding provided crucial economic support that mitigated the potential disruptions simulated in our analysis. The model represents a scenario assuming no government intervention, designed to evaluate economic efficiency losses by holding policy conditions constant. In addition, Baltimore's strong pre-collapse labor market, characterized by high participation rates and low unemployment, also provided greater resilience than the model assumes. Our modeling, therefore, represents a hypothetical worst-case scenario useful for understanding the scale of potential economic disruption in the

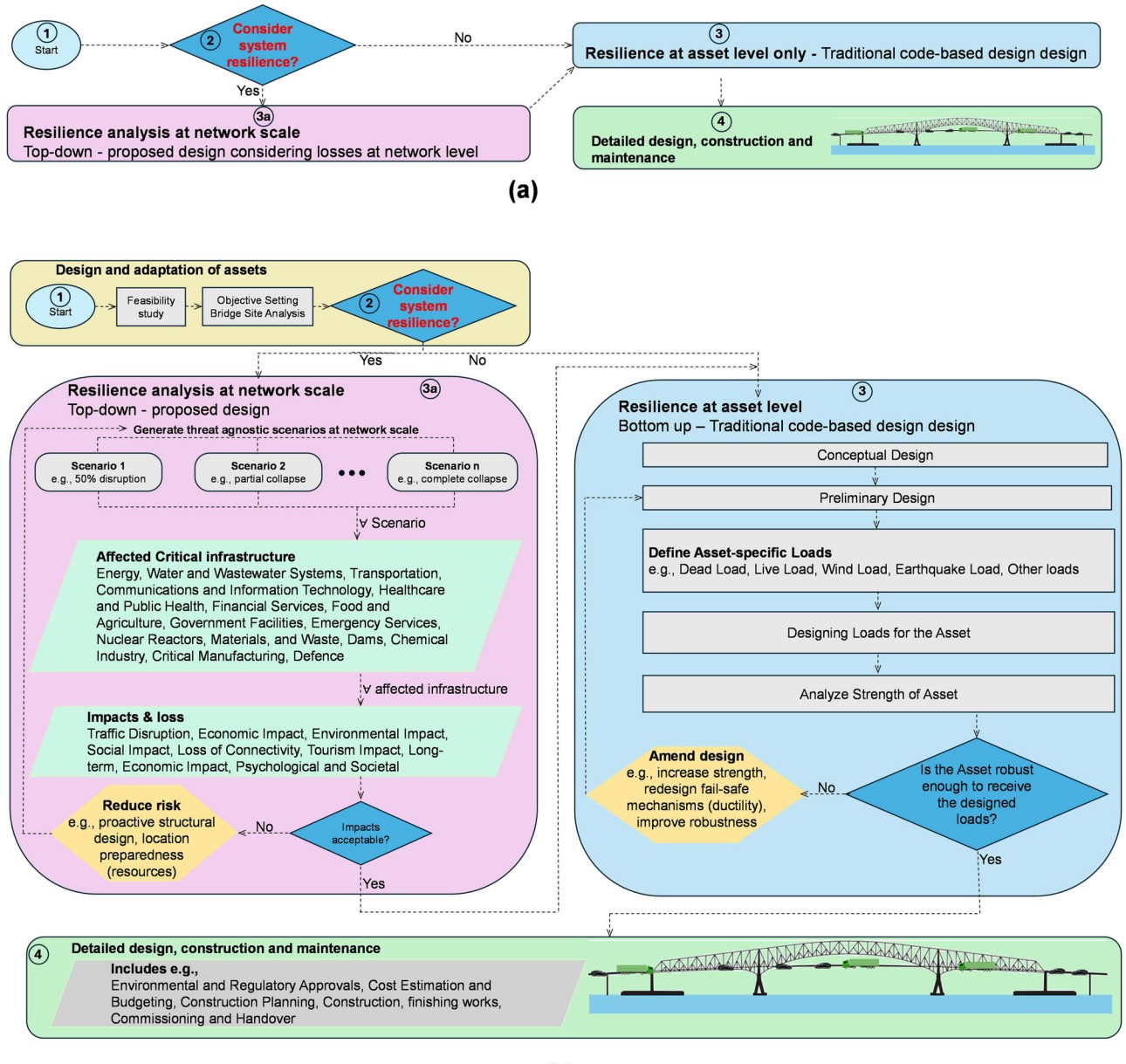

**Fig. 6 | Resilience-Based Bridge Analysis Procedure. a** simplified bridge analysis procedure illustrating a top-down resilience perspective that considers the bridges role in the wider network and (**b**) detailed step-by-step procedure beginning with establishing the bridge's role within broader infrastructure systems (e.g., transportation, energy, water, or economic networks), followed by scenario-based analyses of shutdowns or failures, evaluation of cascading risks and interdependencies, and the development of appropriate mitigation strategies.

absence of mitigation efforts. This approach serves planning and preparedness purposes by quantifying the economic value of rapid policy intervention and infrastructure resilience investments.

Our proposed resilience-based design framework, shown in Fig. 6, is designed to ensure that bridges are modeled and therefore designed as part of a larger system. Figure 6a shows a more simplified version of the process in Fig. 6b. The process begins by determining whether the infrastructure asset (such as a bridge) is designed or assessed as an asset that serves a system or critical network. This step (3a) requires an assessment of the bridge's role within a broader infrastructure context —whether it directly impacts transportation networks, energy distribution, water systems, economic structures, etc. Our case study of the Francis Scott Key Bridge, therefore, only represents a fraction of what may be needed for highly interconnected bridges. This approach is a departure from the known traditional load-based design, which is

also described in Fig. 6. It aims to enhance the network resilience, in addition to the traditional code-based design and assessment requirements to carry traffic and other loads, therefore offering panoramic lenses to structural design as opposed to asset-focused heuristic design.

Once the network context is established, the next step is to simulate different shutdown or failure scenarios for varying lengths of time. This includes partial closures, full shutdowns, and the failure of interconnected systems. For each scenario, we assess the cascading impacts on related infrastructure systems—whether transportation, energy, water, or others—and measure the degree of interdependency. This analysis helps to identify the infrastructure systems that are greatly impacted and require immediate attention. If the cascading risks are deemed unacceptable (i.e., they lead to widespread or critical failures), mitigation strategies must be put in place.

Alternatively, if the analysis reveals minimal interconnection with other critical systems, the focus should shift to enhancing the resilience of the bridge itself. At the asset level, this involves improving structural resilience to withstand external pressures—whether from increased traffic, environmental factors, or unforeseen events. Key design adjustments, such as improving material ductility or adding protective structures, can prevent progressive collapse and ensure long-term resilience.

Our results show that critical infrastructure should be designed and evaluated not only for structural integrity but also for infrastructure geo-economic importance and potential impact of prolonged recovery within interconnected transportation systems. In such an approach, a bridge is seen as an integral asset within a complex system, with its potential failure posing a substantial impact to the overall health and operations of transportation, supply chain, and social networks, and hence to the local and regional economy. This requires collaborative design approaches that establish minimum resilience criteria for interconnected systems.

The resilience-based design framework presented in Fig. 6 represents a shift from traditional load-based approaches to a systems-thinking methodology that considers bridges within their broader infrastructure context. This framework emphasizes the critical importance of assessing cascading impacts and regional interdependencies before determining appropriate mitigation strategies, whether at the network or asset level.

In this paper, we have integrated resilience quantification and regional economic modeling to examine the resilience of the Baltimore region to cascading infrastructure failures. While our analysis focused one specific example, the Key Bridge collapse serves as a watershed moment for infrastructure planning, in general, demonstrating that the traditional focus on load-based risk assessments is insufficient. Future infrastructure must be designed, monitored, and maintained based on resilience and with a full awareness of its role within larger systems. Only through this comprehensive approach can we build truly resilient infrastructure networks capable of withstanding and recovering from both predicted and unforeseen challenges.

Future work can advance a better understanding of the drivers of regional resilience to compounding failures by comparing the resilience of infrastructure systems in different regions to the same kind of disruption, such as a bridge collapse or port closure. While this paper focused on the Francis Scott Bridge, future work should compare the connectivity of bridges nationally or internationally to compare the interconnectedness and resilience of regional economies to infrastructure disruptions. Furthermore, while our analyses focused on infrastructure connected to bridge disruptions, future work will include extending the analyses and approach detailed above to systemic breakdowns in other infrastructure systems, such as energy, water, and supply chains, to aid decision makers in the design and planning of resilient systems.

## Methods
This paper joins two independently developed and documented models to assess the economic impacts of resilience in transportation networks: (1) TranSight[29], a regional economic forecasting model oriented specifically for simulating the outcomes of changes in transportation systems, and (2) the resilience matrix methodology by Linkov et al.[30] to assess the recovery of interconnected bridge and port systems.

### Economic model TranSight
Regional Economic Model, Inc's (REMI)[31,32], TranSight model[29] is designed to be used with transportation forecasting models to translate the outcomes of improvement measures into regional and national economic implications. Regional Economic Models Inc.

(REMI), the developer of TranSight and related products, maintains models for a wide variety of U.S. regions and states in order to support research. For a particular city or region, the effects of transportation projects are forecasted in economic terms that include gross domestic product (GDP), employment, delivered price, commodity access, labor access, and relative cost of production. REMI mixes techniques from Input-Output (I-O) and Computable General Equilibrium (CGE) modeling, as well as economic geography and econometric techniques[33]. The CGE model is designed to simulate how a regional economy finds a new equilibrium after a disruption. However, unless directly included in the inputs, this type of model does not reflect policy responses like state or federal aid. So, in this case, the model represents a scenario of long-term economic outcomes assuming no government intervention. REMI's CGE model assumes general equilibrium across markets and is designed to evaluate the economic efficiency loss due to a disruption by holding other conditions constant.

In this research, TranSight was configured to the Baltimore Metro Area and surrounding regions to assess the Francis Scott Key Bridge and/or Port of Baltimore shutdown scenarios. TranSight was utilized at a local, state, and multi-state level. The model incorporates information on linkages between output and demand, labor and capital demand, population and labor supply, compensation and prices (and costs), and market shares (domestic versus international). The model is U.S.-based (i.e., modeling only economic activity that occurs within the United States) and is run at the national level. The current version of TranSight is calibrated using data through 2023.

The inputs to TranSight from a transportation model include: (a) changes to Vehicle Miles Traveled (VMT), (b) changes to Vehicle Hours Traveled (VHT), and (c) vehicle trips attributable to improvement measures. Changes in velocity (measured by VMT divided by VHT) from the baseline scenario to the improvement (test) scenario are formulated as having a proportional effect on transportation cost, and changes in trips that can be made in a given amount of time (measured by the number of trips divided by VHT) are formulated as having a proportional effect on accessibility cost. Changes from the baseline transportation network are presumed to affect various economic variables via changes in "effective distance," which functions to change travel time or commuting time and expenses. Cost savings due to reduced travel times accrue to industry firms in the model due to reduced commuting and transportation costs and increased access to markets.

Cost-savings, capital investment, and other financial and economic concerns associated with prospective infrastructure projects are related to the regional economy. Changes to economic variables are modeled via policy variables that represent the effect of travel time on individual spending on fuel, and subsequently, disposable income and consumer spending. In addition, costs to industries are related to transportation via the extent to which labor demand can be met and the composite price of goods sent to market. The relationships between these policy variables is presented in Fig. 7. All of the linkages between travel times and associated costs to the policy variables that are used in TranSight are detailed in Model Equations[33] and the TranSight documentation[29].

### Disruption scenarios
In order to examine the cascading and compounding nature of bridge disaster, we look at how the damages due to a bridge shutdown compounds as the length of the closure lengthens, as well as how the two concurrent shutdowns such as a bridge and port closure result in compounding economic effects.

Figure 8 presents considered bridge shutdown-scenarios, each reflecting various levels of operability and structural damage that produce proportionate short-term consequences and often disproportionately larger long-term impacts due to the compounding interdependencies with other infrastructure systems. There are four

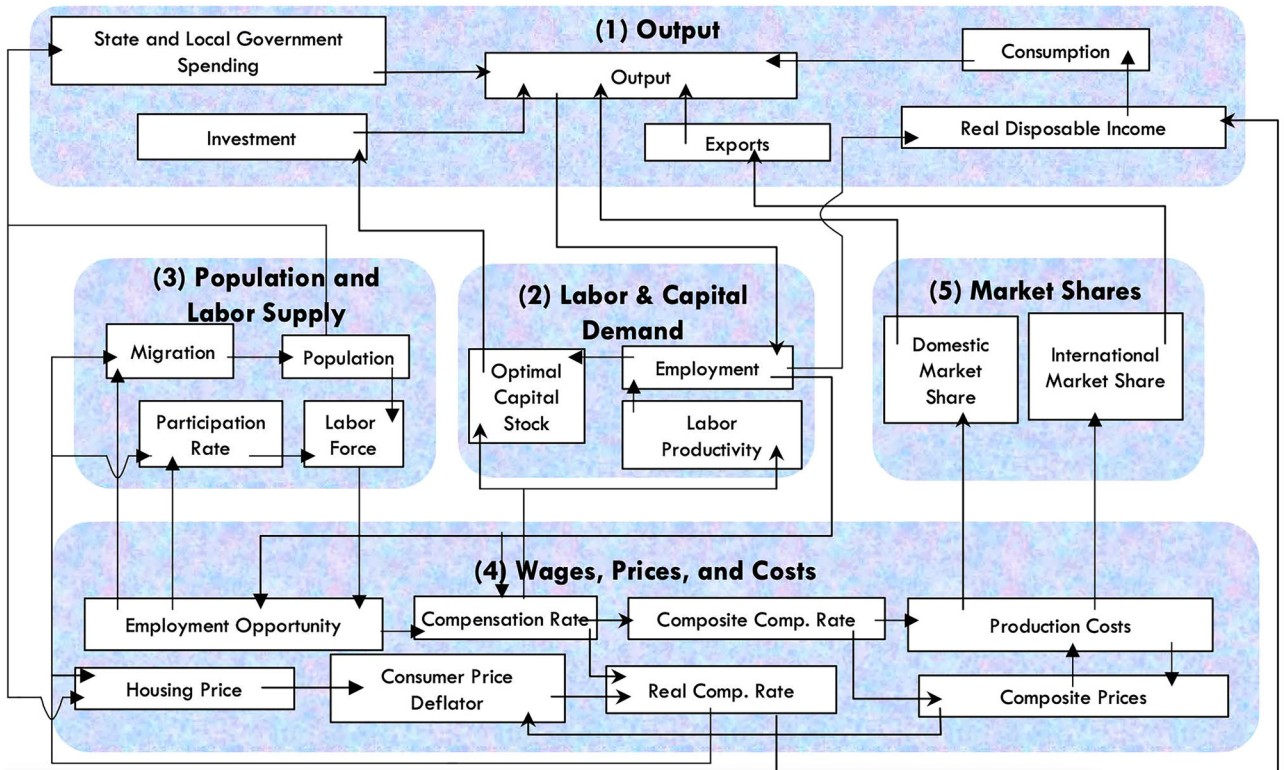

**Fig. 7 | Economic model structure.** The model consists of thousands of simultaneous equations. The overall structure of the model can be summarized in five major blocks: (1) Output and Demand, (2) Labor and Capital Demand, (3) Population and Labor Supply, (4) Compensation, Prices, and Costs, and (5) Market Shares. This Figure shows the blocks and their relationships.

bridge scenarios: normal operation, partial closure, partial damage, and complete damage. Normal operation is the baseline, where the bridge is fully functional with no disruption. Partial closure occurs when the bridge is still open but has reduced capacity, often due to minor issues like debris or spalling, causing localized disruptions.

Our simulations focus on partial damage (1–11 week closure) and complete damage (1–2 year closure). This is because the partial and complete damage scenarios in Fig. 8 involve more bigger disruptions, leading to longer-term economic ripple effects. Furthermore, in the partial and complete damage cases, the bridge closure may cause cascading effects, such as blocking access to a port. In total, 5 simulations were conducted using the TranSight model to compare the impacts of bridge closures, port closures, and combined bridge and port closures of varying length, where the impacts of the closures were project until 2040. A table of the scenarios and their description are given in Table 1.

### Resilience score
The resilience definition used in this paper is based on well-established concepts in the literature. First, the use of the area under the recovery curve as a resilience metric has been used in the past by many authors, and the reader is referred to refs. 34–36 for more information. Second, the normalization by the worst-case scenario when measuring economic resilience is supported by the work of Rose[37], who highlights the importance of using theoretical upper bounds or historical worst-case events as a reference point for resilience measurement. By defining resilience as the complement of the normalized area above the recovery curve, the proposed formulation aligns with the common interpretation of resilience as the ability to withstand and recover from disruptions, ensuring the metric ranges from 0 (complete failure) to 1 (perfect resilience).

Mathematically, let $Q_i(t)$ represent the time-series of the disruption's effect on the economic variable under the disruption scenario $i \in$ {Bridge, Port, Bridge and Port}, and where $t_0$ is the time of the disruptive event and $T$ is the control time. The control time $T$ is determined by the stakeholder and is dependent upon what one is interested in measuring short-term, medium-term, or long-term resilience. For each economic variable of interest, we define the resilience (R) as:

$$R = 1 - \frac{\int_{t_0}^{T} Q_i(t)dt}{T \times Q_i^{worst}} \qquad (1)$$

where $Q_i^{worst}$ is the minimum value of $Q_i(t)$ over the time interval $[t_0, T]$ and all scenarios $i \in$ {Bridge, Port, Bridge and Port}. Or, mathematically this can be written as:

$$Q_i^{worst} = \min_{i, t}\{Q_i(t)\} \qquad (2)$$

This resilience metric represents the complement of the normalized area above the recovery curve, where a value of 1 indicates perfect resilience (i.e., $Q(t) = Q_i^{worst}$ for all $t$) and a value of 0 indicates complete failure (i.e., $Q(t) = 0$ for all $t$). $Q(t)$ in our case usually ranges from $[Q_i^{worst}, 0]$, since it is the effect of a bridge shutdown, which has a negative effect. Positive $Q(t)$ values indicate that the disruption has resulted in better outcomes compared to the baseline no-disruption scenario. A geometric representation of this definition is given in Fig. 9.

### Compounding impacts
To analyze the interaction effects of concurrent infrastructure disruptions, such as a bridge and port closure, we compared two scenarios: (1) a 11-week simultaneous shutdown of both bridge and port

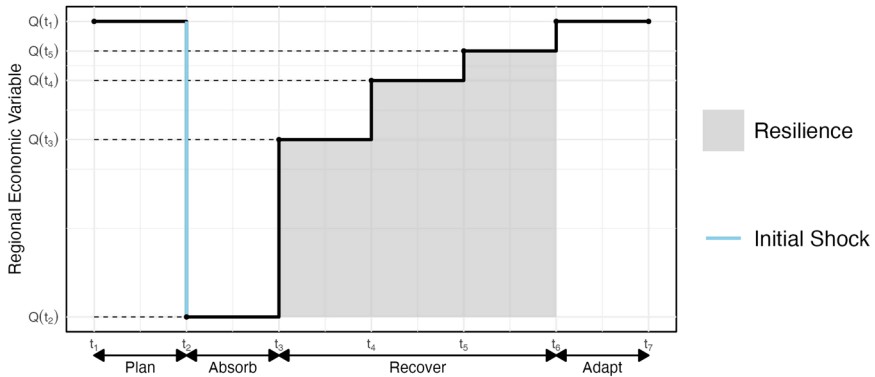

| scenarios of bridge closure for consequence analysis | Structural condition | Operability | Restoration time | Impacts and consequences | | | |
|---|---|---|---|---|---|---|---|
| | | | | Interdependencies with other systems | Society | Economy | Environmental |
| normal operation | no damage, intact | full operability | NA | NA | NA | NA | NA |
| partial closure | minor or no damage e.g., debris, spalling of concrete | road traffic reduced by e.g. 50%, max weight and speed restrictions | Short e.g. 1 to 3 days | accessibility, transport of goods and parts | nuisance, disruption of normal life | supply chain, businesses | increased CO2 emissions, fuel consumption, noise due to traffic detour of heavy lorries |
| partial damage | moderate damage, e.g. failure of one pier or foundation | closed to traffic | medium to long, e.g. 1 to 3 months | accessibility, transport of goods and parts, partial or complete disruption to other networks, e.g. ports, airports, railways islands | loss of culture, reputational, quality of life, claims, political government functions, disruption of emergency services | supply chain, businesses, insurance | increased CO2 emissions, fuel consumption, noise due to traffic detour, contamination due to bridge debris, impacts to ecosystem |
| complete damage | partial or full collapse e.g. deck falls-off piers | closed to traffic | long e.g. 1 to 2 years | | | | |

**Fig. 8 | Scenario comparisons under varying conditions of bridge operation, ranging from normal function to partial closure, partial damage, and complete damage.** Each scenario simulates potential outcomes for medium and long-term infrastructure shutdowns, assessing systemwide economic impacts due to different levels of bridge damage. In this study, partial and complete damage scenarios are explored to understand the compounding effects of prolonged shutdowns on regional economic stability and recovery.

## Table 1 | Simulated Scenarios for Bridge and Port Closures

| Scenario | Description |
|---|---|
| 11-Week Bridge Closure | Temporary bridge closure; quick replacement to allow traffic flow. |
| 11-Week Port Closure | Port closed, disrupting shipping and transport for 11 weeks. |
| 4-Year Bridge Closure | An extended bridge closure with no immediate alternative, causing prolonged disruption. |
| 11-Week Bridge and Port Closure | Simultaneous 11-week closure of both bridge and port, testing resilience to dual disruption. |
| 4-Year Bridge and 11-Week Port Closure | Extended bridge closure with port closed for 11 weeks, testing compounding effects over time. |

**Fig. 9 | A geometric representation of our resilience metric.** The shaded gray area is the resilience. In the above, $Q(t)$ is a measure of regional economic functionality over time, such as the GDP, and shows the variable's response to a shock at time $t_2$ (e.g., a bridge collapse).

facilities, and (3) a 11-week port shutdown paired with a four-year bridge shutdown as outlined in Table 1. These scenarios capture the range of potential impacts across short-term and extended duration disruptions caused by bridge damage.

We quantify the interaction effect by calculating the percent deviation from an additive impact model using the metric:

$$\lambda(t) = \frac{B(t) + P(t) - BP(t)}{|B(t) + P(t)|} \quad (3)$$

where $B(t)$ is the impact of a bridge shutdown, $P(t)$ the port shutdown impact, and $BP(t)$ the combined impact at time $t$. A $\lambda(t)$ of zero represents an additive effect, while non-zero values indicate interaction effects—positive values suggesting a compounded impact, and negative values a mitigative effect.

## Data availability
The economic data used to produce the findings in this study are proprietary and not publicly available. However, they are available from the corresponding author upon request and with permission of Regional Economic Models, Inc.

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

## Acknowledgements

We thank the anonymous reviewers for their constructive feedback, which helped improve the manuscript. We also thank Regional Economic Models, Inc. for their collaboration on this project. Any opinions, findings, and conclusions or recommendations expressed in this material are those of the author(s) and do not necessarily reflect the views of the US Army Corps of Engineers or other funding agencies.

## Author contributions

I.L. and S.M. developed the ideas and approaches, including the resilience-based design framework. S.D. led the paper preparation and interpretation, as well as the modeling of resilience and compounding impacts. B.L., J.D., E.T., O.K., S.G., and A.K. provided REMI modeling. A.B. contributed to the ideas and paper preparation. All authors contributed to the writing and revision of the manuscript and approved the final version.

## Competing interests

The authors declare no competing interests.
