## [Transparent Peer Review File · Nature Communications]

Rethinking infrastructure design from component failure to systemic resilience

Corresponding Author: Dr Igor Linkov

Version 0:

Reviewer comments:

Reviewer #1

(Remarks to the Author)
See Attached Report

Reviewer #2

(Remarks to the Author)

Thanks for the opportunity to review "From Isolated Failures to Systemic Breakdown: Rethinking Infrastructure Design After the Key Bridge Collapse" for Nature Communications.

This paper proposes a systems approach to the design of critical infrastructure that incorporates a resilience and risk approach to understanding recovery and failure. The argues that using this approach will help understand and respond to the potential of cascading failures using the Francis Scott Key Bridge as a case. Using TranSight, a regional economic model, the paper assesses the cascading failure and the economic impact of the collapse and its related disruptions.

I recommend this paper for publication and ask that the authors address these key items:

- The writing is clear in this paper, yet some of the longer sentences detract from the flow of the paper. For example, line 66: "It is not only critical that bridge design move from risk toward resilience ..." should be split into perhaps 2 or more sentences. And line 68: "While probabilistic and large-span bridge traffic load models ..." should also be split. Both sentences use "which" as a transition that complicates the continuity.
- Can you unpack this sentence and explain what you mean by not all bridges should be built equally? I don't disagree, but it's unclear what you're suggesting this at this point. Line 80: "Clearly, not all bridges should be built equal, and so ..."
- In the Economic Impacts section, the paper details several scenarios of job loss. Not all jobs are created equally. Which jobs are these? Are some of these jobs related to just using the bridge? Or are they port labor force related? If port related, can you infer the impacts based on what we know about port worker pay, that they often are at the lower end of the labor force? How might that unevenly impact the region and/or fuel economic disparity?
- Please use color consistency in graphics. Figure 3 is phenomenal. The red and blue colors should be same in Figures 1 and 2 for the 1yr bridge+3mo port and 3mo bridge+3mo port. Then use different colors for the other scenarios in Figures 1 and 2.
- The Discussion and Conclusion section is too long as-is. To be clear the content is excellent, but I'd strongly recommend separating out the Discussion (some of which almost reads as a results section) and the Conclusion. Line 230 could be a Conclusion section.
- The formatting around Figure 4 is hard to follow.
- Please propose the model (Figure 4) in the Discussion, then refer back to it in the Conclusion.
- I see no methodological issues with the economic modeling, kudos.

Version 1:

Reviewer comments:

Reviewer #1

(Remarks to the Author)

Revisions to Address Reviewers' Comments

The authors would like to thank the reviewers and editor for their time and constructive comments, as well as for the opportunity to further improve the quality of the manuscript. Responses and their corresponding revisions are below, line numbers were added to the manuscript to make it easier for reviewers to find changes in the manuscript, where applicable.

Responses Reviewer #1

Reviewer Comment	Response	Line Numbers (If Applicable)
Firstly, the authors use an example that happened last year with scenario analyses and simulated data, but do not explore an event study with actual realized data.	The authors thank the reviewer for pointing this out and agree that it is best to use the actual timelines. When the original set of simulations were done, it was unclear the exact timeline of bridge and port closure. In the revised version, all the simulations have been redone to match more closely the actual timelines of bridge/port shutdowns in Baltimore. Furthermore, discussion has been added and plots of the real data available so far have been provided to create a comparison, and also discuss the role of federal aid and the shortcomings of the model itself.	
Secondly, the authors explore several hypothetical scenarios for one system (Key Bridge and Port of Baltimore), but other infrastructure evaluation exercises (e.g. a rural bridge's impact on its encompassing county's economy, Ambassador Bridge Collapse in Detroit) could help illustrate lower regional economic effects vs. higher regional effects to examine	The authors appreciate this opportunity to explore this point further, and the resulting exploration this comment inspired. However, in response to the other comments by the reviewer, we chose to focus this paper on FSK bridge to allow a comparison to the real-world impact, at least in so far as the impacts have played out thus far.	

how this would alter their respective proposed designs.	In response to this comment, the authors examined bridges in rural areas of Maryland, but from these realized the difficulty in demonstrating a non-result of no effect. For these bridges the expected impact is zero, and therefore do not show much. Truly rural bridges, like the Roddy Shod bridge in MD, for example, would have no impact in the model. It is for this reason that a truly rural, non-interconnected bridge was ultimately not added by the authors. The authors agree a comparison to higher connectivity bridge would be interesting to examine further. However, this paper's focus on bridge-port effects makes a bridge such as Ambassador Bridge, which does not block port access like FSK, difficult to make a straightforward comparison to. So, while the authors appreciate this comment and as a result have begun work on providing different points of comparison, they leave it to a future paper and maintain the focus of this paper on the FSK bridge.	
The authors spend a large portion of the article addressing hypothetical scenarios related to the Key Bridge Collapse, but do not spend much time explaining the realized scenario (i.e. 11-	The authors thank the reviewer for this feedback and have added information and discussion on the actual bridge collapse, as well as how and why the model results are different from the	

week port closure and 4+ year bridge closure) and how it impacted the Baltimore metro economy.	actual bridge collapse. As mentioned above, as well, all the simulations were reperformed to better match the actual timelines.	
The addition of a case study would be useful here in comparing how the Baltimore metro economy evolved compared to their various forecasts as well as a discussion of the various government programs that prevented economic harm (e.g. MD State Legislature’s PORT Act, federal intervention).	The authors agree that the role of various government interventions is a highly valuable case study and have added discussion of how the model does not capture these interventions and how the real impacts have played out in comparison. Economic data for labor force and employment are now included. Various analyses were done to try and get the real effect of the bridge collapse ranging from basic linear models to some ARIMA models. These were deemed to add a little too much to the paper length and perhaps deserve a paper in its own right. Nevertheless, it is clear from the economic data alone that the actual impacts were less pronounced.	
The reality of the matter is that the Bridge collapse had very little noticeable impact on the Baltimore metro economy, according to current labor market data from the Bureau of Labor Statistics. See charts below comparing labor force, employment, and unemployment rate of Baltimore MSA to Philly, Norfolk, and DC. The lack of an effect is likely in part due to government intervention. Additionally, it’s possible that	The authors thank the reviewer for this comment. In short, the model represents a scenario of a long-term economic outcome, holding other conditions constant. Federal aid, an exogenous offset to market outcomes, is not included as a result. From a planning perspective, this highlights the critical role that swift and significant interventions played. This is a crucial point, and so the authors thank the reviewer once again for this comment.	

future revisions to this MSA data may better pick up the effects of the bridge collapse. However, I think at least an acknowledgment of the realized behavior of the economy so far would add value to the manuscript.		
On page 2, the authors state “In situations when interdependency is low, such as bridges in rural areas, a failure might result only in direct repair and replacement costs. Yet, within highly interdependent infrastructure systems inside or near densely populated areas, the scale and scope of distribution may be much higher.” I believe that a regional economic impact evaluation of a rural bridge failure would be a helpful counterpoint to the Key Bridge Collapse in building your argument that these types of evaluations should inform infrastructure design. It might also be helpful to use a third example that uses a more interconnected piece of infrastructure’s failure and its regional impact (e.g. maybe the Ambassador bridge in Detroit qualifies as more interconnected than the Key Bridge, given its role in U.S. trade with Canada).	The authors appreciate this suggestion greatly. As discussed above, ultimately after exploring some additional bridges to include and after including more information comparing the actual outcome, the authors have chosen to focus on the FSK bridge in this paper and having an empirical grounding. As mentioned previously, however, we think this is a critical idea to explore more thoroughly in future work. Various bridges to compare to which may prove more connective, in some sense, including the Ambassador and Golden Gate bridge were considered. This is mentioned as a direction for future work in the paper.	
The addition of the two suggested examples could better motivate the use of your Resilience-Based Bridge Analysis Procedure in Figure 4. The rural bridge example would likely show minimal		

connection with other critical systems, the Key Bridge would show moderate interconnection, and the Ambassador bridge (or another example scenario that you choose) would show high interconnection. The differing resilience scores across the three scenarios would show how infrastructure design planning would be impacted and better generalize your results.		
There is very little difference between the impacts of the 3-month bridge and 3-month port scenario vs. the 3-month port scenarios results across the 5 dependent variables. I would consider dropping the 3-month port closure scenario and explaining the relatively small difference between those two in a footnote. It will make the analysis a bit easier to follow and the graphs easier to read. The essential takeaway is that the 3-month port closure is driving the compounding negative effects in the model, regardless of how long the bridge is shut.	The authors appreciate the opportunity to improve the quality of the manuscript by incorporating this point. IN the Figure caption, it is now mentioned that the 11-week port and then the 11-week port and bridge closure scenario are dominated by the port closure's influence.	

Responses Reviewer #2

Reviewer Comment	Response	Line Numbers (If Applicable)
- The writing is clear in this paper, yet some of the longer sentences detract from the flow of the paper. For example, line 66: "It is not only critical that bridge design move from risk toward	We thank the reviewer for these comments, and agree the language was awkward in some places. Instances of overly complex sentences, such as those mentioned by	

resilience ..." should be split into perhaps 2 or more sentences. And line 68: "While probabilistic and large-span bridge traffic load models ..." should also be split. Both sentences use "which" as a transition that complicates the continuity.	the reviewer, have now been split up or deleted.	
- Can you unpack this sentence and explain what you mean by not all bridges should be built equally? I don't disagree, but it's unclear what why you're suggesting this at this point. Line 80: "Clearly, not all bridges should be built equal, and so ... "	The authors appreciate the opportunity to discuss this point further. The statement "Clearly, not all bridges should be built equal", while perhaps being somewhat provocative, emphasized the core argument of the paper. Existing AASHTO guidelines focus primarily on the probability of physical risk to the bridge. The authors think this is certainly worthwhile and necessary, but that more focus needs to shift away from risk and physical damage and towards resilience and interconnectedness. Two bridges may have the same or similar annual frequency of collapses but should not necessarily be designed to the same standard. Or, more provocatively, they should not be 'built equally'. If the bridge is a critical node in an interconnected system whereas the other is not, then different bridge standards should be applied.	
- In the Economic Impacts section, the paper details several scenarios of job loss.	The authors agree including which industries were impacted is a critical point,	

Not all jobs are created equally. Which jobs are these? Are some of these jobs related to just using the bridge? Or are they port labor force related? If port related, can you infer the impacts based on what we know about port worker pay, that they often are at the lower end of the labor force? How might that unevenly impact the region and/or fuel economic disparity?	and thank the reviewer for pointing this out. As a result, this information has now been included for 7 industries from the North American Industries Classification. Port related labor force falls under “Support Activities for Transportation”, and as the reviewer suggests, this industry was far more negatively impacted in comparison, followed by construction. This information, allow with some discussion of if the impact are nonlinear or not in industry specific job forces are not included in the paper.	
- Please use color consistency in graphics. Figure 3 is phenomenal. The red and blue colors should be same in Figures 1 and 2 for the 1yr bridge+3mo port and 3mo bridge+3mo port. Then use different colors for the other scenarios in Figures 1 and 2.	The authors agree that the previous color scheme was inconsistent and have changed the figures so that the colors are consistent throughout the plots.	
- The Discussion and Conclusion section is to long as-is. To be clear the content is excellent, but I'd strongly recommend separating out the Discussion (some of which almost reads as a results section) and the Conclusion. Line 230 could be a Conclusion section.	Thank you for this helpful structural feedback. We have separated the Discussion and Conclusion sections as suggested, with the new Conclusion beginning at the former line 230. This creates better balance and clarity.	
- The formatting around Figure 4 is hard to follow.	The authors appreciate this comment and have reformatted the Figure accordingly and added some additional description, as well. A more simply formatted version of the step-	

	by-step procedure has been added now, which hopefully makes the framework steps more clear.	
- Please propose the model (Figure 4) in the Discussion, then refer back to it in the Conclusion.	We appreciate this suggestion and have moved Figure 4's presentation to the Discussion section, with appropriate references added in the Conclusion as recommended.	
I see no methodological issues with the economic modeling, kudos.	Thank you for the positive feedback on our economic modeling approach. We're pleased it meets the methodological standards.	

Referee Report on “From Isolated Failures to Systemic Breakdown: Rethinking Infrastructure Design After the Key Bridge Collapse”

Summary

The manuscript addresses the regional economic impact of the Key Bridge Collapse through the analysis of several different scenarios, and how a resilience-based framework should consider the compounding economic effects of infrastructure collapse in the design process. The authors estimate each scenario (e.g. bridge closure, bridge and port closure, port closure) for several time horizons with REMI's Transight model which forecasts changes in several economic variables for the Baltimore area such as GDP and employment. The authors argue that resilience assessments for infrastructure design should account for system-wide effects of disruptions to the wider economy.

Analysis and Recommendation

As it stands, the manuscript outlines a straightforward simulation analysis of the Key Bridge collapse under various scenarios, therefore illustrating the systemic implications of infrastructure disruption on the greater Baltimore economy. While I agree with the authors' approach to informing infrastructure design with these economic shocks and that the Key Bridge collapse is a good motivating example, I feel that the authors could improve upon this manuscript in several ways. Firstly, the authors use an example that happened last year with scenario analyses and simulated data, but do not explore an event study with actual realized data. Secondly, the authors explore several hypothetical scenarios for one system (Key Bridge and Port of Baltimore), but other infrastructure evaluation exercises (e.g. a rural bridge's impact on its encompassing county's economy, Ambassador Bridge Collapse in Detroit) could help illustrate lower regional economic effects vs. higher regional effects to examine how this would alter their respective proposed designs. I believe that the link between regional economic impact analysis and how it informs the resilience-based bridge analysis could be enhanced with the addition of these two examples. Therefore, I recommend a revise and resubmit with a major revision.

Comments

- The authors spend a large portion of the article addressing hypothetical scenarios related to the Key Bridge Collapse, but do not spend much time explaining the realized scenario (i.e. 11-week port closure and 4+ year bridge closure) and how it impacted the Baltimore metro economy. The addition of a case study would be useful here in comparing how the Baltimore metro economy evolved compared to their various forecasts as well as a discussion of the various government programs that prevented economic harm (e.g. MD State Legislature's PORT Act, federal intervention).
- The reality of the matter is that the Bridge collapse had very little noticeable impact on the Baltimore metro economy, according to current labor market data from the Bureau of Labor Statistics. See charts below comparing labor force, employment, and unemployment rate of Baltimore MSA to Philly, Norfolk, and DC. The lack of an effect is likely in part due to government intervention. Additionally, it's possible that future revisions to this MSA data may better pick up the effects of the bridge collapse. However, I think at least an acknowledgment of the realized behavior of the economy so far would add value to the manuscript. Otherwise, why even use the Key Bridge Collapse as a motivating example? There are plenty of hypothetical infrastructure failures that could be simulated within a regional economy forecasting model.

[Figure Redacted]

[Figure Redacted]

[Figure Redacted]

- On page 2, the authors state “In situations when interdependency is low, such as bridges in rural areas, a failure might result only in direct repair and replacement costs. Yet, within highly interdependent infrastructure systems inside or near densely populated areas, the scale and scope of distribution may be much higher.” I believe that a regional economic impact evaluation of a rural bridge failure would be a helpful counterpoint to the Key Bridge Collapse in building your argument that these types of evaluations should inform infrastructure design. It might also be helpful to use a third example that uses a more interconnected piece of infrastructure’s failure and its regional impact (e.g. maybe the Ambassador bridge in Detroit qualifies as more interconnected than the Key Bridge, given its role in U.S. trade with Canada).
- The addition of the two suggested examples could better motivate the use of your Resilience-Based Bridge Analysis Procedure in Figure 4. The rural bridge example would likely show minimal connection with other critical systems, the Key Bridge would show moderate interconnection, and the Ambassador bridge (or another example scenario that you choose) would show high interconnection. The differing resilience scores across the three scenarios would show how infrastructure design planning would be impacted and better generalize your results.
- There is very little difference between the impacts of the 3-month bridge and 3-month port scenario vs. the 3-month port scenarios results across the 5 dependent variables. I would consider dropping the 3-month port closure scenario and explaining the relatively small difference between those two in a footnote. It will make the analysis a bit easier to follow and the graphs easier to read. The essential

takeaway is that the 3-month port closure is driving the compounding negative effects in the model, regardless of how long the bridge is shut.